# Transcriptome Analysis Shows That IFN-I Treatment and Concurrent SAV3 Infection Enriches MHC-I Antigen Processing and Presentation Pathways in Atlantic Salmon-Derived Macrophage/Dendritic Cells

**DOI:** 10.3390/v11050464

**Published:** 2019-05-22

**Authors:** Cheng Xu, Øystein Evensen, Hetron Mweemba Munang’andu

**Affiliations:** Department of Basic Sciences and Aquatic Medicine, Faculty of Veterinary Medicine, Norwegian University of Life Sciences, P.O. Box 369, NO-0102 Oslo, Norway; cheng.xu@nmbu.no (C.X.); oystein.evensen@nmbu.no (Ø.E.)

**Keywords:** antigen, dendritic cells, IFN-I, macrophages, MHC, SAV3, TO-cells

## Abstract

Type I interferons (IFNs) have been shown to play an important role in shaping adaptive immune responses in addition to their antiviral properties in immune cells. To gain insight into the impact of IFN-I-induced pathways involved in early adaptive immune responses, i.e., antigen-presenting pathways, in an Atlantic salmon-derived (*Salmo salar* L.) macrophage cell line (TO-cells), we used a comparative de novo transcriptome analysis where cells were treated with IFN-I or kept untreated and concurrently infected with salmonid alphavirus subtype 3 (SAV3). We found that concurrent treatment of TO-cells with IFN-I and SAV3 infection (SAV3/IFN^+^) significantly enriched the major histocompatibility complex class I (MHC-I) pathway unlike the non-IFN-I treated TO-cells (SAV3/IFN^−^) that had lower expression levels of MHC-I pathway-related genes. Genes such as the proteasomal activator (*PA28*) and β-2 microglobulin (*β2M*) were only differentially expressed in the SAV3/IFN^+^ cells and not in the SAV3/IFN^−^ cells. MHC-I pathway genes like heat shock protein 90 (*Hsp90*), transporter of antigen associated proteins (*TAPs*) and tapasin had higher expression levels in the SAV3/IFN^+^ cells than in the SAV3/IFN^−^ cells. There were no MHC-II pathway-related genes upregulated in SAV3/IFN^+^-treated cells, and cathepsin S linked to the degradation of endosomal antigens in the MHC-II pathway was downregulated in the SAV3/IFN^−^ cells. Overall, our findings show that concurrent IFN-I treatment of TO-cells and SAV3 infection enriched gene expression linked to the MHC-I antigen presentation pathway. Data presented indicate a role of type I IFNs in strengthening antigen processing and presentation that may facilitate activation particularly of CD8+ T-cell responses following SAV3 infection, while SAV3 infection alone downplayed MHC-II pathways.

## 1. Introduction

Since their discovery more than 50 years ago [1], interferons (IFNs) have for a long time only been known for their antiviral properties. Recent advances show that type I IFNs play an important role in modulating adaptive immune responses by playing a pivotal role in the differentiation of circulating monocytes into mature dendritic cells (DCs) and macrophages [1,2,3,4]. They have also been shown to promote the activation of naive CD8 T-cells into cytotoxic T-lymphocytes (CTLs), the polarization of T-helper (Th) cells into Th1 responses, and cross-priming of CD8+ T-cells through the stimulation of DCs [4,5,6,7,8,9]. As pointed out by Gessani et al. [10], the regulatory effects of type I IFNs on T-cell responses stem from their activation of antigen processing in proteasomal compartments, transportation of processed peptides to the endoplasmic reticulum (ER), packaging of processed peptides on major histocompatibility complex (*MHC*) molecules, and finally, presentation of MHC-peptide complexes to naive T-cells. To do these things, type I IFNs up- or downregulate several genes that include MHC-I and II molecules, transport antigen associated proteins (*TAPs*), tapasin, proteasomal activator protein 28 (*PA28*), heat shock proteins (*Hsps*), gamma IFN inducible lysosomal thiol (*GILT*), and several other genes involved in antigen presentation [11,12,13,14,15,16]. In vivo studies on IFN-deficient mice confirmed the crucial role played by endogenous type I IFN in regulating MHC-I peptide complex formation and Hsp70 expression in DCs [17] while Lattanzi et al. [12] have shown that IFNα increases the capacity of DCs to produce high levels of TAP1/2 and tapasin via the modulation of proteasomal activity. These studies show that IFNs regulate the functions of antigen-presenting cells (APCs), such as macrophages and DCs, leading to the activation of T-cell responses [9,10,18,19].

Hence, in the present study we wanted to determine the profile of genes linked to antigen processing and presentation expressed by the macrophage/dendritic-like TO-cells, which are derived from Atlantic salmon head kidney leukocytes, infected with salmonid alphavirus subtype 3 (SAV3) using a comparative de novo transcriptome analysis. Macrophages and dendritic-like cells have been shown to play an important role in antigen presentation in fish and are widely distributed both in lymphoid and mucosal organs in finfish. By comparing the repertoire of genes expressed in SAV3/IFN^+^ TO-cells with those expressed in SAV3/IFN^−^ cells, we wanted to determine the role of IFN-I in regulating the expression of antigen processing and presentation genes concurrently with SAV3 infection in TO-cells.

## 2. Materials and Methods

### 2.1. Cell Culture, Virus Infection, and IFN-I Treatment

TO-cells derived from Atlantic salmon head kidney leukocytes characterized to possess macrophage/dendritic cell-like properties [20,21], were propagated at 20 °C in HMEM (Eagle’s minimal essential medium (MEM]) with Hanks’ balanced salt solution (BSS)) supplemented with L-glutamine, MEM nonessential amino acids, gentamicin sulfate, and 10% Fetal bovine serum. When the cells were 80% confluent, one batch was inoculated with 1 MOI (multiplicity of infection) SAV3 (GenBank accession JQ799139) [22] in triplicates while another batch was treated with 500 ng/mL of Atlantic salmon type I IFN [22] and was concurrently infected with SAV3 in triplicates. Thereafter, the SAV3-infected cells with and without IFN-I treatment were incubated at 15 °C in HMEM maintenance media supplemented with 2% FBS. The mock cells (TO_mock_) were non-infected and only treated with maintenance medium. Cells were harvested after 48 hours and used for total RNA extraction for RNA-seq and qRT-PCR analysis.

### 2.2. Total RNA Isolation

Extraction of total RNA was carried out using the RNeasy Mini Kit with on-column DNase treatment according to the manufacturers’ instructions (Qiagen, Hilden, Germany). The quality and concentration of RNA was analyzed using the ND1000 nanodrop (NanoDrop Technologies, Wilmington, DE, USA) and Agilent 2100 Bioanalyzer (Agilent Technologies, Santa Clara, CA, USA).

### 2.3. Library Construction, Sequencing, and Data Analysis for RNA-Seq

Library construction was carried out by pooling together triplicate samples obtained from total RNA extracted from IFN-treated TO-cells concurrently infected by SAV3. Library construction was also carried out by pooling together a triplicate of total RNA from non-IFN-treated cells infected with SAV3 plus TO_mock_ cells (non-treated/non-infected) for RNA-seq. Treatment of total RNA with DNase I to degrade any possible DNA contamination, enrichment using oligo(dT) magnetic beads, fragmented into approximately 200 bp fragments, synthesis of first-strand cDNA using random hexamer primers followed by synthesis of the second strand together with end reparation coupled with 3′-end single nucleotide A (adenine) addition, ligation of sequence adaptors to the fragments, and fragment enrichment by PCR amplification were also carried out as previously described in our studies [23]. Thereafter, a quality check (QC step) was carried out using the Agilent 2100 Bioanalyzer and ABI StepOnePlus Real-Time PCR System (Waltham, MA, USA) to qualify and quantify the sample library. Subsequently, library products were used for RNA sequencing using Illumina HiSeq^TM^ 2000 (San Diego, CA, USA), BGI-Hong Kong (Hong Kong, China) and clean reads were obtained after the removal of adaptor sequences together with reads having >10% of unknown bases and reads with low-quality bases (base with quality value ≤ 5) >50% in a read.

### 2.4. De Novo Assembly, Functional Annotation, and Gene Ontology Classification

Once a library of clean reads was prepared, clean reads were then used for de novo transcriptome assembly using the Trinity software (http://trinityrnaseq.sourceforge.net/) [24]. Thereafter, the assembled unigenes were annotated into different functional classifications after searching in different protein databases using the BlastX (version 2.2.23) alignment. The four public protein databases used comprised (i) National Center for Biotechnology Information (NCBI) non-redundant (NR), (ii) Swiss-Prot, (iii) Kyoto Encyclopedia of Genes and Genomes (KEGG), and (iv) Cluster of Orthologous Groups (COG) at *e*-value < 0.00001. The direction of the identified unigenes was determined by using the consensus sequence obtained from the four databases [23]. In the case of conflicting results between different databases, the priority order (i) NR, (ii) Swiss-Prot, (iii) KEGG, and (iv) COG was used. BlastX data were used to extract the coding regions (CDS) from unigene sequences and translate them into peptide sequences. Unigenes not identified by BlastX were analyzed using ESTScan to predict their CDS and to decide their sequence direction, while unigenes with NR annotation were further analyzed with Blast2GO (http://www.blast2go.org/) to obtain their gene ontology (GO) annotations. The identified unigenes were classified according to GO functions using the Web Gene Ontology (WEGO) annotation software.

### 2.5. Identification of Differentially Expressed Genes

Mapped read counts for each gene generated from the functional annotation above were normalized for RNA length and total read counts in each lane using the reads per kilobase per million method (RPKM). As such, the RPKM method allowed for direct comparison of the number of transcripts between IFN-treated and non-treated groups, which created the basis for identifying the differentially expressed genes (DEGs). We set the cutoff limit at 95% confidence interval for all RPKM values for each gene and used a rigorous algorithm to generate DEGs by comparing RPKM-mapped reads from IFN-treated and non-treated cells infected with SAV3 versus TO_mock_ cells. Only DEGs with a threshold of false discovery rate (FDR) < 0.001 and an absolute value log_2_ratio > 1 were considered differentially expressed. Thereafter, all identified DEGs were mapped to GO annotations using the Blast2GO software (http://www.blast2go.org/) and were later assigned KEGG ortholog (KOs) identifiers for pathway analysis using the KEGG database (http://www.genome.ad.jp/kegg/). All comparisons were for TO_mock_ against TO infected with SAV3, with or without IFN-1 treatment, and the results are reported as SAV3/IFN^+^ or SAV3/IFN^−^.

### 2.6. Validation of RNA-Seq Data

In order to confirm the validity of our RNA-seq data, eight randomly selected DEGs shown to be up- or downregulated by RNA-seq were used for quantitative real-time PCR (qRT-PCR) analysis using the QuantiFast SYBR Green RT-PCR Kit (Qiagen) and the LightCycler 480 system (Roche, Basel, Switzerland). For each gene, the quantity of template, master mix final volume, reverse transcriptase, PCR initiation activation, and cycles used per reaction were carried out as previously described [23]. Primer sequences used for qRT-PCR are shown in Table 1. The specificity of each PCR product from each primer pair was confirmed by melting curve analysis and agarose gel analysis while the 2^−ΔΔCt^ method was used to quantify the fold increase in gene expression levels relative to the control group. All quantifications were normalized using the β-actin endogenous gene.

## 3. Results

### 3.1. KEGG Pathway Analysis

Table 2 shows that there are 41 unigenes belonging to the antigen processing and presentation pathway differentially expressed in the SAV3-infected cells while 35 were differentially expressed in the SAV3/IFN^+^ group, accounting for 0.44% and 2.15% of the annotated DEGs for each group, respectively. Table 2 shows that the antigen processing and presentation pathway designated as K04612 by KEGG analysis was significantly enriched (*p* = 3.2132e−09) for the type I IFN-treated cells (SAV3/IFN^+^) and not enriched (*p* = 0.9999) for the non-IFN-treated (SAV3/IFN^−^) indicating that concurrent IFN-I treatment and SAV3 infection significantly enriched the antigen processing and presentation pathways in these cells. As such, the proportion of upregulated unigenes in the SAV3/IFN^+^ cells (91.43%, *n* = 35) was higher than that in the SAV3/IFN^−^ cells (41.46%, *n* = 41). As shown in Table 3 and Figure 1, there were more downregulated unigenes in the SAV3/IFN^−^ cells than in the SAV3/IFN^+^ cells. In order to better understand which components of the antigen processing and presentation pathway were involved, we investigated the gene profiles of the MHC-I and -II pathways expressed in SAV3/IFN^+^ and SAV3/IFN^−^ cells.

### 3.2. Major Histocompatibility Complex I Pathway

Genes differentially expressed in response to SAV3 infection with or without IFN-I treatment, belonging to the MHC-I pathways were broadly classified into two groups, the proteasome and endoplasmic reticulum expressed genes.

#### 3.2.1. Proteasome Genes

Among the proteasomal degradation pathway-related genes, proteasomal activator 28 (PA28) proteins was upregulated in the SAV3/IFN^+^ and not in the SAV3/IFN^−^. Heat shock protein 70 (Hsp70) was downregulated in the SAV3/IFN^−^ while Hsp90 was upregulated in both SAV3/IFN^+^ and SAV3/IFN^−^ (Table 3). However, expression levels of Hsp90 were higher in the SAV3/IFN^+^ cells (Table 4).

#### 3.2.2. Endoplasmic Reticulum Genes

Genes differentially expressed linked to the endoplasmic reticulum (ER) included the transport associated proteins 1 (*TAP1*) and 2 (*TAP2*) as well as the TAP binding protein (*TAPBP*), which is also called tapasin (Table 3). All these genes were upregulated in both the SAV3/IFN^+^ and SAV3/IFN^−^ and expression levels were highest in the SAV3/IFN^+^ cells (Table 4). In addition, β-2 microglobulin (*β2M*), was not expressed in SAV3/IFN^−^ and was upregulated in SAV3/IFN^+^. On the contrary, calnexin (*CANX*) was downregulated in SAV3/IFN^−^. The network pathways linking proteasomal differentially expressed genes with ER genes in the MHC-I pathway are shown in Figure 2A,B. It is noteworthy, that there were 12 unigenes classified as MHC-I and all were upregulated in the SAV3/IFN^+^ cells. On the contrary, only two unigenes were classified as MHC-I in the SAV3/IFN^−^ cells of which one was upregulated while the other was down regulated. Overall, Figure 1 and Table 3 show that the number of unigenes upregulated in the SAV3/IFN^+^ cells was higher than that in the SAV3/IFN^−^ cells. Figure 1 and Table 3 also show that the number of downregulated unigenes was higher in the SAV3/IFN^−^ than that in the SAV3/IFN^+^ cells. Moreover, Table 4 shows that the fold increase in expression levels was higher in the SAV3/IFN^+^ cells than in the SAV3/IFN^−^ cells.

### 3.3. Major Histocompatibility Complex II Pathway

In general, Figure 2A,B shows that there were fewer genes differentially expressed in response to SAV3 infection in TO-cells belonging to the MHC-II pathway for both SAV3/IFN^+^ and SAV3/IFN^−^ cells, and, of those differentially regulated, the genes were broadly classified into the endosomal and endolysosomal compartments.

#### 3.3.1. Endosomal Compartment Genes

In the endosomal compartment, only the gamma IFN inducible lysosomal thiol gene (*GILT*) was downregulated in SAV3/IFN^+^ while no endosomal gene was differentially expressed in SAV3/IFN^−^ (Table 4 and Figure 2B).

#### 3.3.2. Endolysosomal Compartment Genes

In the endolysosomal compartment, only three cathepsin (*cat*) genes were differentially expressed in SAV3/IFN^−^, which included catL (*CTSL*) and CatS (*CTSS*). Among these, CTSL was upregulated while CTSS was downregulated (Table 4). Both Table 4 and Figure 2B show that there were no endolysosomal compartment genes expressed in the SAV3/IFN^+^ cells. In summary, only the SAV3/IFN^−^ had upregulated genes (*catL*) while the SAV3/IFN^+^ had no upregulated genes.

### 3.4. Other Genes

Other genes differentially expressed in response to SAV3 infection in TO-cells (SAV3/IFN^−^) linked to antigen processing and presentation included the regulatory factor X (*RFX*), cyclic AMP (adenosine monophosphate) response element binding protein (*CREB*), PAX (Paired box)-interacting protein 1 (*PAXIP1*), recyclin-1 (*RCY-1*), filaggrin (*FLG*), and nuclear factor Y (*NFY*) genes, all downregulated (Table 3 and Table 4). Other genes downregulated in the SAV3/IFN^+^ cells included protein disulfide isomerase family A, member 3 (*PDIAS3*) (Table 3 and Table 4).

### 3.5. Quantitative Real Time PCR

Figure 3 shows selected genes for qRT-PCR analysis of the antigen presentation pathway genes detected by RNA-seq. Hsp90, TAP1 and 2, MHC-I, and β2M were upregulated in SAV3/IFN^+^ and SAV3/IFN^−^ with expression level higher in the SAV3/IFN^+^. Cathepsin B and cathepsin S were downregulated in the SAV3/IFN^−^ and upregulated in the SAV3/IFN^+^, which conforms to observations from RNA-seq data. MHC-II was not differentially expressed in any groups. Overall, Figure 3 shows that genes that were up- or downregulated by RNA-seq were also up- or downregulated by qRT-PCR thereby confirming the validity of our RNA-seq data.

## 4. Discussion

### 4.1. KEGG Pathway Analysis

Pathway-based analysis of transcriptome data has emerged to be the most effective approach used for identifying network pathways of genes that regulate immune responses in different host cells. Using a comparative pathway analysis, we have shown that concurrent addition of IFN-I with SAV3 infection in TO-cells significantly enriched the antigen processing and presentation pathway for SAV3/IFN^+^ compared to that for SAV3/IFN^−^. As a result there were more genes upregulated linked to the MHC-I pathway in the SAV3/IFN^+^ in contrast to SAV3/IFN^−^. Further comparative analyses of individual genes expressed in the IFN-I-treated cells compared to the non-IFN-treated cells showed that all genes linked to the MHC-I pathway upregulated in the SAV3/IFN^+^ were higher than the levels expressed in SAV3/IFN^−^. These findings show an increase in genes associated with antigen processing and presentation via the MHC-I pathway in SAV3-infected TO-cells treated with IFN prior to infection which is in line with observations made by Lattanzi et al. [12], who showed that IFNα regulates cytosolic antigen processing and presentation via the MHC-I pathway in mouse DCs. Overall, the approach used in this study shows that combining de novo transcriptome assembly with KEGG pathway analysis can be used to identify the network pathway of genes induced by different regulatory cytokines in fish cells with potential APC properties.

### 4.2. MHC-I Pathway Analysis

Intracellular antigen processing is largely dependent on the degradation of cytosolic antigens into peptides using proteolytic enzymes [25]. In mammals, two major proteins that degrade cytosolic antigens into peptides in the proteasome are PA28 and Hsp90 [26,27,28,29,30], which have also been characterized in fish [31,32,33,34]. Yamano et al. [35] have shown that these proteins carry out antigen degradation independent of each other and interestingly Hsp90 was enriched in SAV3/IFN^−^ whereas both Hsp90 and PA28 were enriched in SAV3/IFN^+^ cells. Yamano et al. [35] also pointed out that apart from playing a major role in antigen degradation, Hsp90 serves as a post proteasomal peptide carrier that delivers the processed peptides to the TAPs to prevent degradation of the processed epitopes by cytosolic peptidases [36]. It is likely that the SAV3/IFN^+^ that had higher Hsp90 levels had a corresponding higher capacity of transporting its peptides into the ER than the SAV3/IFN^−^, which could account for the high TAP levels together with ER downstream genes detected in the SAV3/IFN^+^ compared to SAV3/IFN^−^.

Once viral antigens are degraded into peptides in the proteasome, the processed peptides are transported into the lumen of the ER by the TAP transporters. In the ER, the peptides are transferred from the TAP transporters onto MHC-I molecules. The MHC-I molecules that bind to processed peptides are made of two chains namely the polymorphic MHC-1α chain and non-polymorphic β2M chain, which assemble in the ER to form the MHC-1α/β2M/peptide trimers [37,38]. Peaper and Cresswell [37] pointed out that formation of stable MHC-1α/β2M/peptide is largely dependent on tapasin, which is responsible for optimal loading of the peptides onto the MHC-1α/β2M molecules. Additionally, calnexin (CANX) promotes the early folding of MHC-I heavy chains, while calreticulin (CALR) and ERp57 are involved in peptide loading onto the MHC-I molecules [37]. 

In this study, only tapasin was upregulated in both treatment/infection groups while CALR, ERp57, and CANX were either downregulated or not differentially expressed. It is interesting to note that β2M was only expressed in the SAV3/IFN^+^ cells, suggesting that the IFN-I-treated cells had higher capacity to form MHC-1α/β2M/peptide complexes. In addition, the high levels of MHC-I and tapasin expressed in the SAV3/IFN^+^ suggest that this group potentially had a higher capacity of forming stable MHC-Iα/β2M/peptide complexes than the SAV3/IFN^−^. These findings are in agreement with Landis et al. [39] who showed upregulation of β2M and tapasin in rainbow trout macrophages infected by viral hemorrhagic septicemia virus (VHSV) and Hansen and La Patra [40] who showed β2M and MHC-I upregulation in infectious hematopoietic necrosis virus (IHNV)-infected cells, which coincided with type I IFN upregulation. Jorgensen et al. [41,42] showed MHC-I and β2M upregulation, which corresponded with upregulated of IFN-I when MHC-II was downregulated in Atlantic salmon infected with infectious salmon anemia virus (ISAV). They observed that this phenomenon was more pronounced in the head kidney and not in other organs of infected fish. They also noted that CD8α was upregulated during the viremia stage when CD4 was not differentially expressed. In this study, we found that genes linked to CD8+ T-cell activation were upregulated, while genes linked to CD4+ T-cell activation were not differentially expressed in SAV3/IFN^+^. Overall, these findings suggest that IFN-I enhances SAV3 antigen processing and presentation via the MHC-I pathway.

### 4.3. MHC-II Pathway Analysis

Antigen processing in the endosome involves the degradation of the endocytosed antigens using proteases [43]. For MHC-II molecules to bind the processed peptides, they must gain access into the endosomal MHC-II compartments (MIIC). To do so, newly synthesized MHC-II molecules bind to the non-polymorphic protein called the invariant chain (Ii or CD74) to form Ii-MHC-II complexes. The Ii protein contains targeting motifs that direct the MHC-II molecules to the MIICs [44]. Once in the MIICs, MHC-II molecules bound to Ii cannot bind the processed antigenic peptides until they have been proteolytically degraded and dissociated from the Ii-MHC-II complexes [45,46]. Degradation of the Ii-MHC-II complex is carried out in a series of steps using different proteins such as Cathepsin B (*CTSB*) until the MHC-II molecule is left with a fragment of Ii called CLIP (class II Ii peptide), which is later replaced with peptides from endosomal degraded antigens [46,47]. Studies on human APCs have shown that the removal of CLIP is facilitated by human leukocyte antigen DM (HLA-DM) [44,46,48], whose interaction with MHC-II also edits the peptides bound to MHC-II by ensuring that only peptides with higher affinity are selected for presentation to CD4 T-cells [49,50]. To date no similar peptide editing system to the human HLA-DM has been characterized in teleosts fish [51]. However, indications are that MHC-II antigen presentation mechanisms found in mammals also exists in fish given that most of the genes linked to the MHC-II pathway found in mammals, inclusive of Ii [52,53] and MHC-II [54,55,56,57], have also been characterized in fish. 

In this study, neither MHC-II nor Ii were differentially expressed in SAV3/IFN^+^ or SAV3/IFN^−^. CTSB was upregulated, which played a significant role in endosomal antigen degradation in the SAV3/IFN^−^, and not in the SAV3/IFN^+^, where the underlying mechanisms are not known or easily explained. Given that TO-cells used in this study were harvested at 48 hours post infection, it remains unknown whether subsequent follow-up beyond 48 hours post infection would have led to upregulation of MHC-II and Ii genes linked to SAV3 presentation via the MHC-II pathway. There is need for follow-up studies with sequential sampling post infection with our without treatment to determine whether SAV3 infections (in the absence of IFN-I) can lead to upregulation of MHC-II and Ii genes in TO-cells. However, the general failure to upregulate genes linked to the MHC-II pathway in SAV3/IFN^+^ further consolidates our notion that IFN-I treatment of TO-cells polarized SAV3 antigen presentation via the MHC-I pathway.

## 5. Conclusions

In this study, we have shown that IFN-I polarizes the upregulation of genes linked to SAV3 antigen processing and presentation via the MHC-I pathway in TO-cells. In addition, we have also shown that by combining comparative transcriptome analysis with network pathway analyses, it is possible to identify the most relevant genes and network pathways regulated by different cytokines in virus-infected fish cells. However, it is important to point out that the analyses carried out here were based on genes generated by de novo assembly, they do not show the functional mechanisms of these genes. Hence, there is need for follow-up studies to demonstrate the exact mechanisms used by these genes to activate the MHC-I pathway following IFN-I treatment and SAV3 infection in TO-cells. Even so, data presented here could serve as a roadmap for elucidating underlying mechanisms mediated by IFN-I to facilitate adaptive immune responses in fish.

## 6. Data Access

The RNA-sequencing data used in this study have been deposited in the National Center for Biotechnology Information (NCBI) Gene Expression Omnibus (GEO) database accession number GSE64095 (www.ncbi.nih.gov/geo Accession number GSE64095).

## Figures and Tables

**Figure 1 viruses-11-00464-f001:**
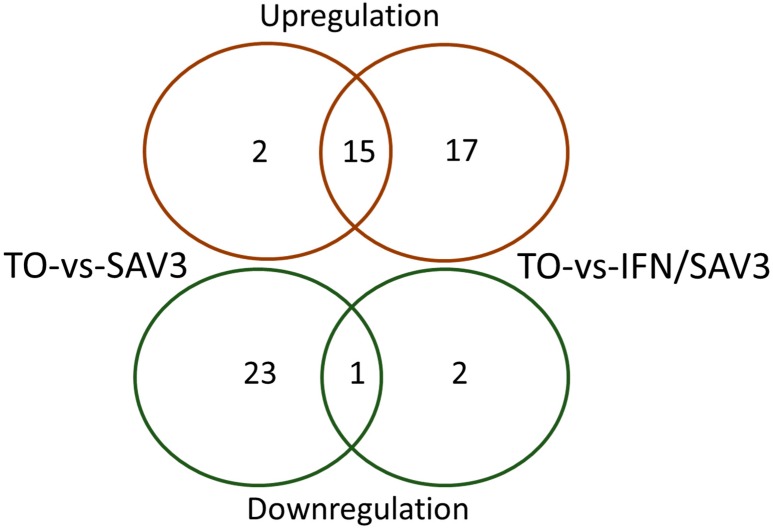
Shows the distribution of upregulated and downregulated unigenes differentially expressed in the TO-vs.-SAV3 and TO-vs.-SAV3/IFN cells involved in the antigen processing and presentation pathway.

**Figure 2 viruses-11-00464-f002:**
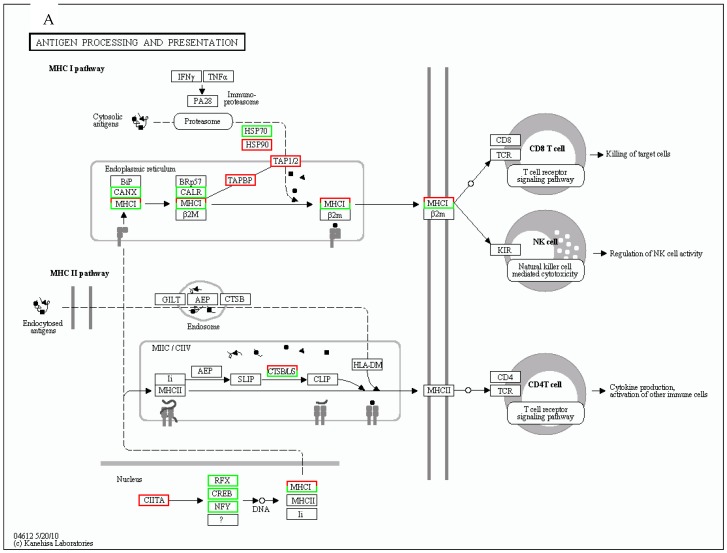
Shows antigen processing and processing pathways for the TO-vs.-SAV3/IFN and TO-vs.-SAV3 generated by the KEGG pathway software analysis. (**A**) Network pathway for TO-vs.-SAV3. (**B**) Network pathway of TO-vs.-SAV3/IFN antigen processing and presentation in TO-cells. Red squares depict upregulated genes, while green squares depict downregulated genes in the pathway. Dots, triangles and squares show broken down antigens as cytosolic antigens in MHC-I and endocytosed antigens in MHC-II pathways. Solid lines show activation and dotted lines show alternative pathways. The mixed shading of red/green squares represents a mixed population of upregulated and downregulated genes. Note that there were more downregulated genes in the type I IFN-treated cells (TO-vs.-SAV3/IFN) (Figure 2A) than in the non-treated cells (TO-vs.-SAV3) (Figure 2B).

**Figure 3 viruses-11-00464-f003:**
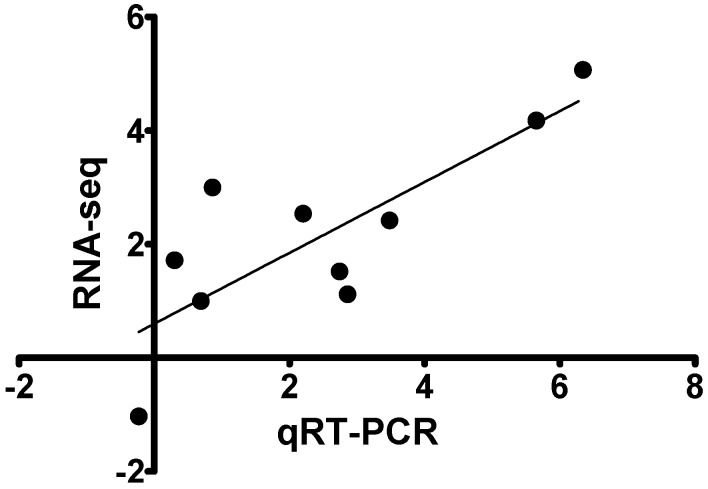
Shows a significant linear correlation (*r* = 0.8027, *p* = 0.0052) between RNA-seq log_2_ fold change and quantitative RT-PCR data for the randomly selected genes generated as the mean of the fold increase in gene expression relative to the TO control group after normalization with beta-actin.

**Table 1 viruses-11-00464-t001:** Primer sequences for real time (RT) PCR.

Primer Name		Sequence
*HSP90*	F	CCACCATGGGCTACATGATG
	R	CCTTCACCGCCTTGTCATTC
*TAP1*	F	ACGAGCCTGAAGCCTTTAC
	R	TCACACACAAACTCACACAC
*TAP2*	F	GGGAAACAGAAGACACAGAAG
	R	ATGCCCCAACCAAAAGGAG
*β2M*	F	TCGTTGTACTTGTGCTCATTTACAGC
	R	CAGGGTATTCTTATCTCCAAAGTTGC
*MHCI*	F	CTGCATTGAGTGGCTGAAGA
	R	GGTGATCTTGTCCGTCTTTC
*MHCII*	F	TCTCCAGTCTGCCCTTCACC
	R	GAACACAGCAGGACCCACAC
*CathepsinS*	F	CGAAGGGAGGTCTGGGAGAGGAAT
	R	GCCCAGGTCATAGGTGTGCATGTC
*CathepsinB*	F	TGTGAGACTGGATACACACCTGGCTAC
	R	GCTCCTTCCACAGGTCCGTTCTTC
*β-actin*	F	CCAGTCCTGCTCACTGAGGC
	R	GGTCTCAAACATGATCTGGGTCA

**Table 2 viruses-11-00464-t002:** Kyoto Encyclopedia of Genes and Genomes (KEGG) pathway analysis of antigen presentation pathway genes expressed in the TO-vs.-SAV3 and TO-vs.-SAV3/IFN.

Parameters	TO-vs.-SAV3	TO-vs.-SAV3/IFN
Antigen presentation pathway ID	K04612	K04612
DEG with pathway annotation of antigen presentation	41 (0.44%)	35 (2.15%)
Total DEG with pathway annotations	9315	1629
All genes with pathway annotation of antigen presentation	145	145
Q-value	1.0000e+00	1.2654e−07
*p*-value	0.9999	3.2132e−09
Number of upregulated unigenes	17	32
Number of downregulated unigenes	24	3
Percentage of upregulated unigenes	41.46%	91.43%
Percentage of downregulated unigenes	58.44%	8.57%

*Note:* SAV3—salmonid alphavirus subtype 3; IFN—inteferon; DEG—differentially expressed gene.

**Table 3 viruses-11-00464-t003:** Summary of number of antigen processing and presentation unigenes expressed in TO-vs.-SAV3/IFN and TO-vs.-SAV3.

Gene Name	Abbrev.	KEGG	Up/Down	TO-vs.-SAV3 (Unigenes)	TO-vs.-SAV3/IFN (Unigenes)
*Cathepsin L*	*CTSL*	K01365	Up	3	
*Class II transactivator*	*CIITA*	K08060	Up	3	3
*Transport associated protein 2*	*TAP2*	K05654	Up	2	4
*Transport associated protein 1*	*TAP1*	K05653	Up	2	2
*TAP binding protein (tapasin)*	*TAPBP*	K08058	Up	2	3
*Heat shock protein 90*	*HSP90*	K04090	Up	5	5
*Major histocompatibility complex I*	*MHC-I*	K06751	Up	1	12
*Lysophosphatidic acid receptor 5*	*LPAR5*	K08390	Up		1
*Proteasome activator 28*	*PA28*	K06697	Up		1
*Beta-2-microglobulin*	*β2M*	K08055	Up		1
*Major histocompatibility complex, class I*	*MHC-I*	K06751	Down	1	
*Cyclic AMP response element-binding protein 1*	*CREB-1*	K05870	Down	5	
*Heat shock protein 70*	*HSP70*	K09489	Down	7	
*PAX-interacting protein 1*	*PAXIP1*	K14972	Down	1	
*Recyclin-1*	*RCY1*	K15071	Down	1	
*Non-specific serine/threonine protein kinase*	*STK*	K08282	Down	1	
*Calreticulin*	*CALR*	K08057	Down	2	1
*Nuclear factor Y*	*NFY*	K08066	Down	1	
*Filaggrin*	*FLG*	K10384	Down	1	
*Regulatory factor X-associated protein*	*RFXAP*	K08063	Down	1	
*Calnexin*	*CANX*	K08054	Down	2	
*Gamma IFN inducible lysosome thiol*	*GILT*	K08059	Down		1
*Protein disulfide isomerase family A, member 3*	*PDIA3*	K08056	Down		1
Total unigenes expressed				41	35

**Table 4 viruses-11-00464-t004:** Comparison of fold increase in expression level of antigen processing and presentation genes expressed in the TO-vs.-SAV3/IFN and TO-vs.-SAV3.

Gene Name	Abbrev.	KEGG	Unigene	GeneBank	Up/Down	TO-vs.-SAV3		TO-vs.-SA-3/IFN	
						Log_2_ Ratio (PD/TO)	*p*-Value	Log_2_ Ratio (PD/TO)	*p*-Value
*Cathepsin L*	*CTSL*	K01365	Unigene20896	BT043962.1|	Up	11.54	2.84E−51		
*Class II transactivator*	*CIITA*	K08060	CL7384.1	NP_001186995.1	Up	9.31	1.22E−17	9.87	9.83E−25
*Transport associated protein 2*	*TAP2*	K05654	CL7652.1	Z83329.1	Up	4.18	1.14E−142	1.38	6.21E−08
*Transport associated protein 1*	*TAP1*	K05653	Unigene6385	AF115538.1	Up	1.52	2.83E−124	2.34	0
*TAP binding protein (tapasin)*	*TAPBP*	K08058	CL1564.1	NM_001124553.1	Up	1.85	3.36E−110	2.01	8.00E−305
*Heat shock protein 90*	*HSP90*	K04090	CL11346. 1	NM_001173702.1	Up	1.72	1.62E−73	3.00	7.42E−182
*Lysophosphatidic acid receptor 5*	*LPAR5*	K08390	Unigene7594	NP_001133983.1	Up			1.50	4.47E−57
*Proteasome activator 28*	*PA28*	K06697	CL10. 2	ACM08763.1	Up			1.15	3.41E−27
*Beta-2-microglobulin*	*β2M*	K08055	Unigene75.1	AF180485	Up			1.12	0
*Major histocompatibility complex, class I*	*MHC-I*	K06751	Unigene8180	L63541.1	Up	1.10	3.14E−10	2.74	7.64E−155
*Major histocompatibility complex, class I*	*MHC-I*	K06751	Unigene23484	BT072706.1	Down	−1.11	3.25E−06		
*Cyclic AMP response element-binding protein 1*	*CREB-1*	K05870	CL3991.2	EMC89054.1	Down	−2.52	7.25E−06		
*Heat shock protein 70*	*HSP70*	K09489	CL1805. 2	NP_999881.1	Down	−1.03	6.00E−80		
*PAX-interacting protein 1*	*PAXIP1*	K14972	CL5737. 1	NM_001025462.1	Down	−1.68	3.83E−07		
*Recyclin-1*	*RCY1*	K15071	CL1550. 1	NM_001165325.1	Down	−1.65	4.03E−39		
*Non-specific serine/threonine protein kinase*	*STK*	K08282	Unigene4701	XM_004075980.1	Down	−1.64	2.57E−14		
*Calreticulin*	*CALR*	K08057	CL1908. 1	XM_004068445.1	Down	−1.91	5.93E−07	−1.33	1.66E−05
*Nuclear factor Y*	*NFY*	K08066	CL10158. 1	AC133690.1	Down	−1.39	1.67E−15		
*Filaggrin*	*FLG*	K10384	CL9396. 1	XM_003437908.1	Down	−1.38	8.33E−29		
*Regulatory factor X*	*RFX*	K08063	Unigene16842	XM_004319280.1	Down	−1.17	5.47E−05		
*Calnexin*	*CANX*	K08054	CL8069. 2	AAQ18011.1	Down	−1.16	4.79E−-54		
*Cathepsin S*	*CTSS*	K01365	Unigene2181	NP_001134963.1	Down	−1.03	1.85E−61		
*Interferon, gamma-inducible protein 30*	*GILT*	K08059	Unigene12822	ACI67174.1	Down			−1.10	6.76E−06
*Protein disulfide isomerase family A, member 3*	*PDIA3*	K08056	Unigene19161	NP_998070.1	Down			−1.05	7.23E−-07

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
