# Peer review of "Transcriptome Analysis Shows That IFN-I Treatment and Concurrent SAV3 Infection Enriches MHC-I Antigen Processing and Presentation Pathways in Atlantic Salmon-Derived Macrophage/Dendritic Cells"

_viruses, 2019, doi:10.3390/v11050464_

Reviewer 1 Report

Interesting piece of work providing further insights on the in vitro role and mechanisms of IFN-1 in TO cells upon infection with SAV-3.

While as authors point out there is still a requirement for demonstration of the mechanisms used by the genes to perform the activation of the MHC-1 pathway, the research it has provided a sound further evidence of the involvement.

Please consider

Lines 65-67 check for clarity explaining content of each experimental group/batchs; although it does become clearer at lines 78-80 it should be so the firdt time.

Line 118 check where the font change begins.

Line 137 suggest a coma after "group", and “accounting” rather than “account”.

Line 151 infection with "or" without” rather than “our”.

Line 160 is probably not intended to be bold font, if is to follow as previous subtitles.

Caption of Fig 1 can be shortened by allocating the 1A or 1B the first time the groups are refer to (currently it is refer to x3 times)

Line 209 missing an “in” after word “expressed”.

Author Response

We greatly appreciate the queries raised by the reviwer. We found them constructive and very helpful in improving the quality of our manuscript. We have answered all queries raised on a point-by-point basis as shown below and we hope in the current state our manuscript will be considered for publication.

QUERY 1) Lines 65-67 check for clarity explaining content of each experimental group/batchs; although it does become clearer at lines 78-80 it should be so the firdt time.

RESPONSE: The sentence “Parallel samples were infected with SAV3 without 
 IFN-I treatment” has been deleted to avoid confusion

QUERY 2) Line 118 check where the font change begins.

RESPONSE: Thanks correction has been done.

QUERY 3) Line 137 suggest a coma after "group", and “accounting” rather than “account”.

RESPONSE: Correction done as suggested , see line 137.

QUERY 4) Line 151 infection with "or" without” rather than “our”.

RESPONSE: Correction done as suggested, see line 151

QUERY 5) Line 160 is probably not intended to be bold font, if is to follow as previous subtitles.

Caption of Fig 1 can be shortened by allocating the 1A or 1B the first time the groups are refer to (currently it is refer to x3 times)

RESPONSE: Correction done as recommended, see line 160.

QUERY 6) Line 209 missing an “in” after word “expressed”.

RESPONSE: Correction done as recommended

Reviewer 2 Report

This manuscript compare the impact of SAV infection versus SAV infection plus type I IFN treatment on the expression of genes involved in Ag presentation and processing, in macrophage like TO cells (Atlantic salmon). The data suggest that the addition of IFN significantly increases the response of the mhc pathway , compared to the viral infection alone. 

A number of issues that need to be addressed: 

Major / general comments

1) Many sentences are not clear. The whole manuscript should be carefully edited.

2) It is really a pity that cells treated with IFN only were not included in the study, even if the transcriptome response of TO cells has been compared by the same group a few years ago. (Xu C, Evensen O, Munang'andu HM (2015) De novo assembly and transcriptome analysis of Atlantic salmon macrophage/dendritic-like TO cells following type I IFN treatment and Salmonid alphavirus subtype-3 infection. BMC Genomics 16: 96. 10.1186/s12864-015-1302-1 [doi];s12864-015-1302-1 ).

3) The results of the differential analyses SAV3/TO and SAV3+IFN/TO should be presented, not only what refers to Ag presentation. It seems that the first analysis identified many more genes differentially expressed than the SAV3+IFN, which is unexpected and should be fully described and explained (line 135). Descriptive statistical analysis and Venn diagrams describing the overlap between the two gene lists should be provided.  This is important to put Table 2 in the context, and to understand what the P values really means for the KEGG ID in each case.

In keeping with this, for the gene analyzed in this study, it would be useful to provide the expression level (ie, counts in each point ). It is unexpected that B2M is not detected at all in SAV3 treated cells.  

4) the designof the experiment is not clear . The section " Cell culture, virus infection and IFN-I treatment 
needs to be clarified. For example line 64: " When the cells were 
80% confluent, one batch was inoculated with 1 MOI SAV3 (Genbank accession JQ799139) [22] in triplicates while another batch was treated with 500 ng/ml of Atlantic salmon type I IFN and was 
 concurrently infected with SAV3 in triplicates. Parallel samples were infected with SAV3 without 
 IFN-I treatment." 

Do the authors mean that one batch was infected by SAV, one infected by SAV and treated with IFN and a third batch only treated by SAV ?

5) The differential analysis is not well described either. Line 99 : what means  "best alignments "? Section " Identification of differentially expressed genes 
: the authors used rpkm, which may not be the best option (see https://www.ncbi.nlm.nih.gov/pmc/articles/PMC4702322/; in table 3, no adj p value of equivalent metric is provided.

6) The authors write line 166 abou mhc 1 pathway " Overall, Figures 1A and 1B  show that there were more upregulated genes in the SAV3/IFN+ than in SAV3/IFN-, and expression levels were also highest in SAV3/IFN+ (Table 3). 
"

This is not what Figure 1A (with IFN)  and B (without IFN) show for the class I pathway !! This should be explained/ clarified.  

Minor points:

Introduction:

Clarify sentences line 48-50 and 52-53

A recent article by Sobhkhez may be cited and discussed

Figure 2: the legend is incomplete. What is shown, FC?  log2 FC?

The correlation, for this type of measures is rather low and raises the issue of correspondence between the mapping of results in rnaseq and the isoforms amplified by PCR.  

Line 223 "These findings indicate that IFN-I increased the antigen processing and presentation capacity of SAV3 infected TO-cells via the MHC-I pathway" is an overstaement

Line 60 . Treatment with salmon type I IFN: the IFN should be clearly identified (for exmaple with an offical  sequence ID). Salmon IFN are diverse and the nomenclature is complicated.

Line 78-80 : please re phrase.

Line 120, line 151, : our should be "or"

Line 155-158 : re phrase and clarify

Line 167: what is "respectively "for ?

Line 195/6: are cathepsins only involved in the mhc class 2 pathway?

Line 203: what means "randomly selected" ?             

Author Response

VIRUSES-465230: RESPONSE TO REVIEWER-1

We greatly appreciate the queries raised by the reviwer. We found them constructive and very helpful in improving the quality of our manuscript. We have answered all queries raised on a point-by-point basis as shown below and we hope in the current state our manuscript will be considered for publication.

QEUERY 1) Many sentences are not clear. The whole manuscript should be carefully edited.

RESPONSE:  We have read through the entire manuscript and made corrects as recommended.

QUERY 3) The results of the differential analyses SAV3/TO and SAV3+IFN/TO should be presented, not only what refers to Ag presentation. It seems that the first analysis identified many more genes differentially expressed than the SAV3+IFN, which is unexpected and should be fully described and explained (line 135). Descriptive statistical analysis and Venn diagrams describing the overlap between the two gene lists should be provided.  This is important to put Table 2 in the context, and to understand what the P values really means for the KEGG ID in each case. In keeping with this, for the gene analyzed in this study, it would be useful to provide the expression level (ie, counts in each point ). It is unexpected that B2M is not detected at all in SAV3 treated cells.  

RESPONSE: We have added two additional Tables (Table 2 and 3) as well as Figure 1 showing a venn diagram as recommended. In Table 4, what was Table 3 in first submission we have added p-values to support the expression reflected as log2 of fold increase. In addition more genes also included as recommended.

QUERY 4) the design of the experiment is not clear . The section " Cell culture, virus infection and IFN-I treatment 
needs to be clarified. For example line 64: " When the cells were 
80% confluent, one batch was inoculated with 1 MOI SAV3 (Genbank accession JQ799139) [22] in triplicates while another batch was treated with 500 ng/ml of Atlantic salmon type I IFN and was 
 concurrently infected with SAV3 in triplicates. Parallel samples were infected with SAV3 without 
 IFN-I treatment." 
Do the authors mean that one batch was infected by SAV, one infected by SAV and treated with IFN and a third batch only treated by SAV ?

RESPONSE: The sentence Parallel samples were infected with SAV3 without 
 IFN-I treatment has been deleted. See lines 65-68

QUERY 5) The differential analysis is not well described either. Line 99 : what means  "best alignments "? Section " Identification of differentially expressed genes 
: the authors used rpkm, which may not be the best option (see https://www.ncbi.nlm.nih.gov/pmc/articles/PMC4702322/; in table 3, no adj p value of equivalent metric is provided.

RESPONSE: ‘Best alignment’ has been deleted and replaced with ‘concensus sequence’. We have also added a reference indicating the method used is described previously in our earlierpublication.

QUERY 6) The authors write line 166 abou mhc 1 pathway " Overall, Figures 1A and 1B 
 show that there were more upregulated genes in the SAV3/IFN+ than in SAV3/IFN-, and expression 
levels were also highest in SAV3/IFN+ (Table 3). 
" This is not what Figure 1A (with IFN)  and B (without IFN) show for the class I pathway !! This should be explained/ clarified.  

RESPONSE: this sentence has been rephrased to indicate that Table 3 and Figure 1 show that there were more ugreulated genes in the SAV3/IFN+ than in the IFN/SAV3- cells, see lines 172-177.

Minor points:

Introduction:

QUERY 7) Clarify sentences line 48-50 and 52-53, A recent article by Sobhkhez may be cited and discussed

RESPONSE: Both sentences have been clarified see lines, see lines 48-50 and lines 52-54.

QUERY 8) Figure 2: the legend is incomplete. What is shown, FC?  log2 FC?

RESPONSE: Figure 2  complete, see lines 235-237

The correlation, for this type of measures is rather low and raises the issue of correspondence between the mapping of results in rnaseq and the isoforms amplified by PCR.  

RESPONSE: FC means fold change and this has been corrected in Figure legend 2, see lines 235-237

QUERY 9) Line 223 "These findings indicate that IFN-I increased the antigen processing and 
presentation capacity of SAV3 infected TO-cells via the MHC-I pathway" is an overstaement 

RESPONSE: Sentences has been rephrased, see lines 248-252

QUERY 10) Line 60 . Treatment with salmon type I IFN: the IFN should be clearly identified (for exmaple with an offical  sequence ID). Salmon IFN are diverse and the nomenclature is complicated.

RESPONSE: A reference of previous publication has been added to this.

QUERY 11) Line 78-80 : please re phrase.

 RESPONSES: Sentence has been rephrased, see lines 78-80

QUERY 12) Line 120, line 151, : our should be "or"

RESPONSES: correction done

QUERY 13) Line 155-158 : re phrase and clarify

RESPONSES: sentence has been modified, see lines 155-159.

QUERY 14) Line 167: what is "respectively "for ?

RESPONSES: The word “respectively” has been deleted from the sentence

QUERY 15) Line 195/6: are cathepsins only involved in the mhc class 2 pathway?

RESPONSES: part of sentence indication cathepsins involvement in MHC-II pathway has been deleted.

QUERY 16) Line 203: what means "randomly selected" ? 

RESPONSES: The word randomly has been deleted, this is already explained in line 123.

Round  2

Reviewer 2 Report

All my comments have been addressed. Please correct the detail below. 

The legend of the new "Figure 1" is 

Shows the distribution of upregulated and downregulated unigenes differentially expressed in the TO-VS-SAV3 and TO-VS-IFN/SAV3 cells"  It must be indicated that the figure shows ONLY genes involved in the antigen processing and presentation, as mentioned for Tables 3 and 4.

Author Response

QUERY: he legend of the new "Figure 1" Shows the distribution of upregulated and downregulated unigenes differentially expressed in the TO-VS-SAV3 and TO-VS-IFN/SAV3 cells"  It must be indicated that the figure shows ONLY genes involved in the antigen processing and presentation, as mentioned for Tables 3 and 4.

RESPONSE: Correction has been made as suggested by the reviewer. See legend for Figure 1, see lines 180-182